# Manipulating hyperbolic transient plasmons in a layered semiconductor

Rao Fu[1,9], Yusong Qu[2,9], Mengfei Xue[3,9] ✉, Xinghui Liu[4], Shengyao Chen[5], Yongqian Zhao[1,6], Runkun Chen[7], Boxuan Li[1], Hongming Weng[1,8], Qian Liu[2,5] ✉, Qing Dai[2] ✉ & Jianing Chen[1,8] ✉

Anisotropic materials with oppositely signed dielectric tensors support hyperbolic polaritons, displaying enhanced electromagnetic localization and directional energy flow. However, the most reported hyperbolic phonon polaritons are difficult to apply for active electro-optical modulations and optoelectronic devices. Here, we report a dynamic topological plasmonic dispersion transition in black phosphorus via photo-induced carrier injection, i.e., transforming the iso-frequency contour from a pristine ellipsoid to a non-equilibrium hyperboloid. Our work also demonstrates the peculiar transient plasmonic properties of the studied layered semiconductor, such as the ultrafast transition, low propagation losses, efficient optical emission from the black phosphorus's edges, and the characterization of different transient plasmon modes. Our results may be relevant for the development of future optoelectronic applications.

Anisotropy has proven to offer exciting possibilities for understanding light-matter interaction processes in optics. Layered materials with positive dielectric tensors, such as $ReS_2$, $MoS_2$, and $SnS$[1-3], are known to preserve anisotropic waveguides. When a material's dielectric tensor exhibits oppositely signed elements, its iso-frequency contours (IFCs) appear with hyperboloid shapes and allow hyperbolic polaritons. Hyperbolic polaritons have been the subject of many studies in layered materials, such as h-BN, $WSe_2$, $ZrSiSe$, $\alpha$-$MoO_3$, $\alpha$-$V_2O_5$, and natural crystals like tin oxide, calcite, $\beta$-$Ga_2O_3$, and $CdWO_4$[4-12], which are highly desirable in various applications[13-17]. Usually, the hyperbolic IFCs in momentum space are closely related to remarkable polaritonic properties, including enhanced electromagnetic field localization and energy flow engineering.

Most hyperbolic phonon polaritons exhibit insusceptibility to electro-optical manipulation. However, layered semiconductors, such as Black phosphorus (BP), may offer an alternative due to the possibility of manipulating the charge carriers. The BP, a typical layered semiconductor with layer-dependent direct bandgap ranging from 1.7 eV in a monolayer to 0.3 eV in bulk[18], has garnered extensive research interest due to its anisotropic band structure. These unique features have promoted BP as a promising platform for developing innovative devices, such as high-performance electronics, infrared sensors, and electronic band engineering[19-23].

The anisotropy in the BP's electronic band and optical permittivity could result in significant anisotropic optical conductivity[24-26], which suggests the BP supports hyperbolic plasmon polaritons[27-30], but its

[1]Beijing National Laboratory for Condensed Matter Physics, Institute of Physics, Chinese Academy of Sciences & School of Physical Sciences, University of Chinese Academy of Sciences, Beijing 100190, China. [2]CAS Key Laboratory of Nanophotonic Materials and Devices, National Center for Nanoscience and Technology & School of Nanoscience and Engineering, University of Chinese Academy of Sciences, Beijing 100190, China. [3]Suzhou Laboratory, Suzhou 215100, China. [4]State Key Laboratory of Quantum Optics and Quantum Optics Devices, Institute of Laser Spectroscopy, Collaborative Innovation Center of Extreme Optics, Shanxi University, Taiyuan, Shanxi 030006, China. [5]MOE Key Laboratory of Weak-Light Nonlinear Photonics, TEDA Institute of Applied Physics, School of Physics, Nankai University, Tianjin 300457, China. [6]Wenzhou Institute, University of Chinese Academy of Sciences, Wenzhou 325001, China. [7]State Key Laboratory of Structural Chemistry, Fujian Institute of Research on the Structure of Matter, Chinese Academy of Sciences, Fuzhou 350002, China. [8]Songshan Lake Materials Laboratory, Dongguan, Guangdong 523808, China. [9]These authors contributed equally: Rao Fu, Yusong Qu, Mengfei Xue. ✉e-mail: xuemf@szlab.ac.cn; liuq@nanoctr.cn; daiq@nanoctr.cn; jnchen@iphy.ac.cn

low intrinsic electrical carrier doping level hinders the practical realization of the hyperbolic plasmons[31–33].

This work activated and manipulated the robust transient hyperbolic plasmons in the BP slabs by inducing the BP's dielectric tensors with opposite signs by ultrafast laser pulses. Usually, the photo-induced electronic transitions can significantly boost carrier density[5,34,35], which offers a practical route for optically tuning plasmons. The photo-pumping process in our work injects ample hot charge carriers into the BP, thereby triggering hyperbolic plasmons that are otherwise absent in the pristine state. These resulting non-equilibrium states then allow the IFCs to topological transition from the pristine ellipsoid to a rare transient hyperboloid, which is distinctive in plasmons[6,36]. Analysis of the dynamic features of the non-equilibrium states at different pump-probe delays revealed the coexistence of a ~5 ps propagating plasmonic mode and a ~40 ps localized edge plasmonic mode.

## Results
### Realization of transient hyperbolic plasmons

Figure 1a shows the experimental setup of the nanoscopy with ultrafast lasers utilized to study the photo-induced hyperbolic plasmonic response of the BP slabs (see Methods). Since the BP is a biaxial crystal, acquiring its non-equilibrium dielectric tensor is necessary to determine the transient plasmonic dispersion. The dielectric tensor of the bulk BP has a diagonal form $[\varepsilon] = diag[\varepsilon_x, \varepsilon_y, \varepsilon_z]$ ($x$ is armchair, $y$ is zigzag); thus, the dielectric elements $\varepsilon_j$ are typically depicted using the Drude expression[37]:

$$\varepsilon_j = \varepsilon_{\infty,j} - \varepsilon_{pl,j} = \varepsilon_{\infty,j} - \frac{\omega_{p,j}^2}{\omega^2 + i\omega\gamma_j}; j = x,y,z \quad (1)$$

Where $\gamma_j$ is the carrier scattering rate, $\omega_{p,j} = \sqrt{\frac{ne^2}{m_j^*\varepsilon_0}}$ is the plasmon frequency, $n$ is the electron or hole density, $\omega$ is the angular frequency of probe light, $\varepsilon_{\infty,j}$ is the high-frequency permittivity, $\varepsilon_0$ is the vacuum permittivity, $m_j^* = \frac{m_{e,j}^* m_{h,j}^*}{m_{e,j}^* + m_{h,j}^*}$ is the reduced effective mass combining the

electron ($m_e^*$) and the hole ($m_h^*$)[34]. Equation 1 indicates that $\varepsilon_j$ is obtained by subtracting the $\varepsilon_{pl,j}$ from the $\varepsilon_{\infty,j}$, which signifies that a positive $\varepsilon_{\infty,j}$ weakens the negative change induced by $\varepsilon_{pl,j}$ in $\varepsilon_j$, the dielectric screening effect[37].

The ultrafast optical pumping (photon energy $h\nu = 0.8$ eV) triggers the electronic transition from the valence band to the conduction band, as shown in Fig. 1b. The dispersion of the electronic bands in the zigzag direction is flatter than the armchair direction, resulting in a larger effective mass[26]. From $m_{(e,h),j}^* = \hbar^2 k_j \frac{\partial k_j}{\partial E_{(C,V)}}$[38], we obtain the anisotropic reduced effective mass, where: $m_x^* = 0.12m_0$, $m_y^* = 0.32m_0$, $m_z^* = 0.10m_0$, $m_0$ is the electron mass. These values can be used to obtain the pump-excited anisotropic plasmonic resonance with the Drude ratio[39,40]:

$$\frac{\omega_{p,y}^2}{\omega_{p,x}^2} = \frac{m_x^*}{m_y^*} = 0.38, \frac{\omega_{p,z}^2}{\omega_{p,x}^2} = \frac{m_x^*}{m_z^*} = 1.2 \quad (2)$$

Figure 1c shows BP's nano-Fourier-transform infrared spectroscopy (nano-FTIR), with a pumping fluence of 0.5 mJ/cm² at $h\nu = 0.8$ eV to inject the photo-induced carriers. We fitted the nano-FTIR spectra using a dipole model incorporating the effective dielectric constant $\langle\varepsilon\rangle = \sqrt{\langle\varepsilon_{plane}\rangle\varepsilon_z}$ at a pump-probe delay $\tau = 200$ fs with a carrier concentration corresponding to $1.3 \times 10^{19}$ cm⁻³. The dielectric tensor of BP was extracted through this method, where $\langle\varepsilon_{plane}\rangle$ is the averaged in-plane dielectric constant[41,42].

Figure 1d presents the real part of dielectric elements by fitting the spectra shown in Fig. 1c. The signs of the real part of dielectric elements in the armchair and zigzag direction are both positive as a result of the intensely screened plasmon frequency $\omega_{p,j}^* = \frac{\omega_{p,j}}{\sqrt{\varepsilon_{\infty,j}}}$ ($\varepsilon_{\infty,x} = 18$ and $\varepsilon_{\infty,y} = 14$). However, the sign of the real part of the dielectric element in the $z$ direction becomes negative due to a much weaker dielectric screening ($\varepsilon_{\infty,z} = 9.75$)[43]. Consequently, the non-equilibrium BP exhibits

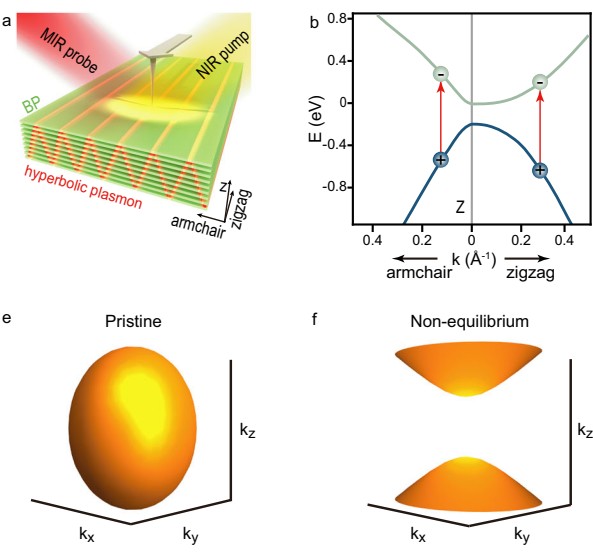

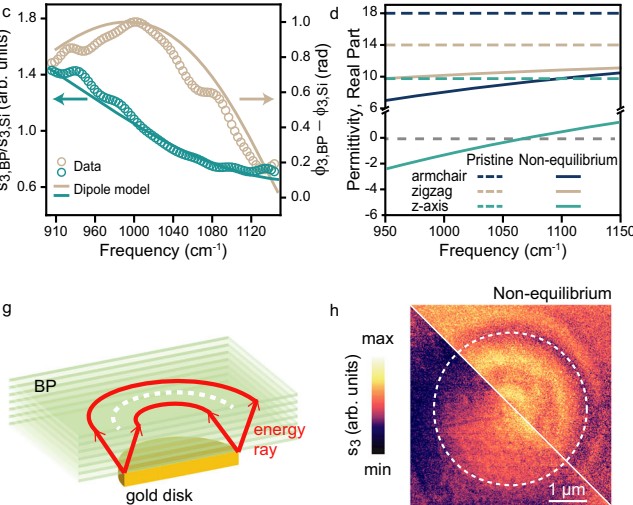

**Fig. 1 | Photo-induced optical anisotropy. a** Schematic diagram of the experimental setup and the transient hyperbolic plasmons. **b** Band structure of bulk BP near Z point in Brillouin region. The red arrows indicate the pump-induced electronic transition. **c** Normalized nano-FTIR amplitude $s_{3,BP}/s_{3,Si}$ (green) and phase $\phi_{3,BP} - \phi_{3,Si}$ (brown) spectra of the pump excited BP at $\tau = 200$ fs with 0.5 mJ/cm² pump fluence. The opened circles represent the experimental data, and the solid lines are from the dipole model calculation. **d** The real part of dielectric elements of the bulk BP. Dashed lines: pristine BP. Solid Lines: BP in non-equilibrium **e**, **f** IFCs of the BP at $\nu/c = 950$ cm⁻¹, where $\nu$ is the frequency of probe light and $c$ is the velocity of light. **e** pristine BP. **f** BP in non-equilibrium. **g** Schematic of the hyperbolic plasmons launched by the edge of the gold disk. The red arrows represent energy rays. **h** The near-field image $s_3$ of pristine and non-equilibrium BP/gold stacked structure at $\tau = 200$ fs with 0.5 mJ/cm² pump fluence. The frequency region of the probe is $\nu/c = 850–1200$ cm⁻¹. The white dashed lines in **g**, **h** mark the dark edge of the gold disk.

oppositely signed dielectric elements, suggesting considerable optical anisotropy between the in and out-of-plane.

To elucidate the hyperbolic dispersion, the IFCs of the BP slabs is employed ($k_j$ is the wavevector along $j$ direction, $j = x, y, z$; $k_O$ is the wavevector of the free space light)[44]:

$$\frac{k_x^2}{\varepsilon_y \varepsilon_z} + \frac{k_y^2}{\varepsilon_x \varepsilon_z} + \frac{k_z^2}{\varepsilon_x \varepsilon_y} = \frac{\varepsilon_x + \varepsilon_y}{2\varepsilon_x \varepsilon_y} k_0^2 \qquad (3)$$

Hyperbolic IFCs at $v/c = 950\,\text{cm}^{-1}$ were realized by substituting these dielectric elements into Eq. (3) with and without pump excitation. The results, presented in Fig. 1e, f, reveal a topological transition of the IFCs from the pristine ellipsoidal shape to a non-equilibrium hyperbolic shape, thus affirming the capability of the BP slabs to activate photonic switching hyperbolic transient plasmons[24,25].

We further positioned a 310-nm-thick BP on a round gold disk to experimentally verify the presence of hyperbolic plasmons. Figure 1g shows that the hyperbolic IFCs allow the gold disk's edge to launch conical-shaped energy rays, producing two bright rings separated by a dark ring above the round gold disk's edge[5,6,13,45]. Accordingly, the near-field image in Fig. 1h of the non-equilibrium state BP/gold stacked structure clearly shows a dark ring sandwiched by two bright rings above the gold disk, while the pristine BP/gold does not exhibit any ring pattern, confirming the excitation of the photo-induced out-of-plane hyperbolic transient plasmons in BP (Supplementary Note 8).

## Propagation mechanism of plasmons

The robust and complex near-field interference patterns allow retrieval of the subtle scattering mechanisms of the BP's transient hyperbolic plasmons. Figure 2a shows the 353-nm-thick BP slab, with the edge parallel to the zigzag direction (Supplementary Note 1). Figure 2b presents the hyperbolic transient plasmonic fringes propagating along the armchair direction of the BP slab. Figure 2c shows the angle of the incidence, $\theta$, defined as the angle between the BP edge and the projection of the probe beam, is varied in this experiment. We found that when the polarization direction of the $p$-polarized probe beam is perpendicular to the BP edge ($\theta = 90°$ and $270°$), the BP edges act as efficient antennas to launch transient plasmons.

Figure 2b and d display two sets of fringes with different fringe spacings. The short-period fringes dominate when the probe beam is perpendicular to the BP edge ($\theta = 90°$ and $270°$), while the long-period fringes dominate when the probe beam is parallel to the BP edge ($\theta = 0°$ and $180°$). The period of the long-period fringes is equal to the transient plasmons wavelength $\lambda_p$[46,47]. Figure 2e shows the amplitude of the Fourier transformation of the fringe curves in Fig. 2d ($\theta = 0°$ and $90°$), which showed the wavevector ratio between the long and short period of fringes is 2, indicating the coexistence of both the tip and edge-launched plasmons.

To describe the $\theta$ dependence of near-field fringes, we employed the following model, taking into account the tip and edge-launching mechanisms and the geometrical decay[48]:

$$E_{total}(x) \propto 1 + \frac{A_1}{\sqrt{x}} e^{i(q_p x + \varphi_1)} + \frac{A_2 |\sin\theta|}{x} e^{i(q_p x + \varphi_2)} + \frac{A_3 |\sin\theta|}{x\sqrt{x}} e^{i(2q_p x + \varphi_3)}$$

$$(4)$$

The $A_1, A_2 |\sin\theta|, A_3 |\sin\theta|$ describe the relative strength of the tip-launching + edge-emission route, the edge-launching + tip-emission

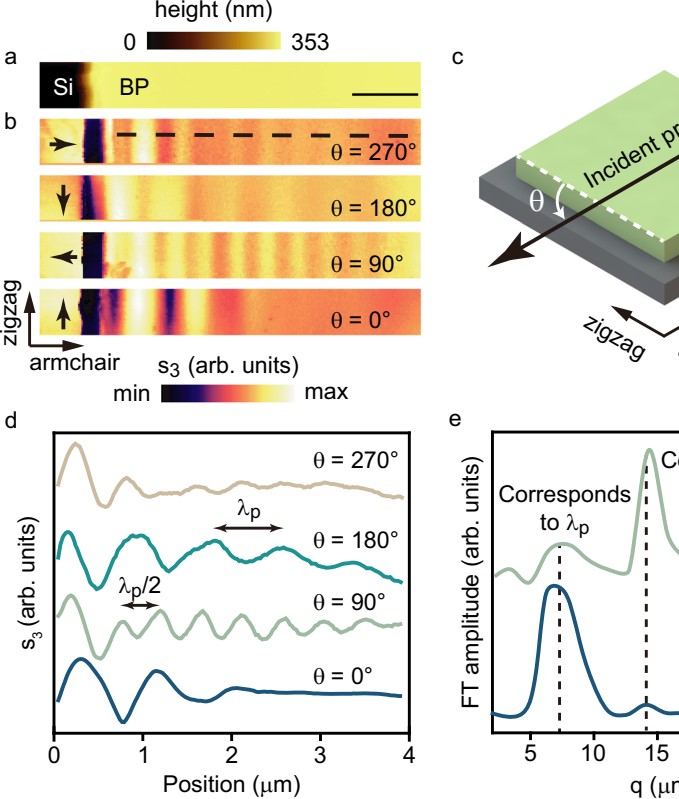

**Fig. 2 | Probe-geometry-dependent plasmonic patterns. a** The topography image of a 353-nm-thick BP slab, the scale bar is 1 μm. **b** Near-field amplitude images $s_3$ of polaritons propagating along the armchair direction with different $\theta$, where the black arrows are incident probe projection directions. The frequency region of the mid-infrared probe is $v/c = 850-1200\,\text{cm}^{-1}$. **c** Schematic diagram of the angle $\theta$ between the projection direction of the incident probe (black arrow) and the zigzag direction (white dashed line). **d** The fringe profiles in lines, which are taken along the dashed line in **b**, shows the existence of plasmon wavelength $\lambda_p$ and half of plasmon wavelength $\lambda_p/2$, respectively. **e** The Fourier transform spectra of fringe curves in **d** at $\theta = 0°$ and $90°$.

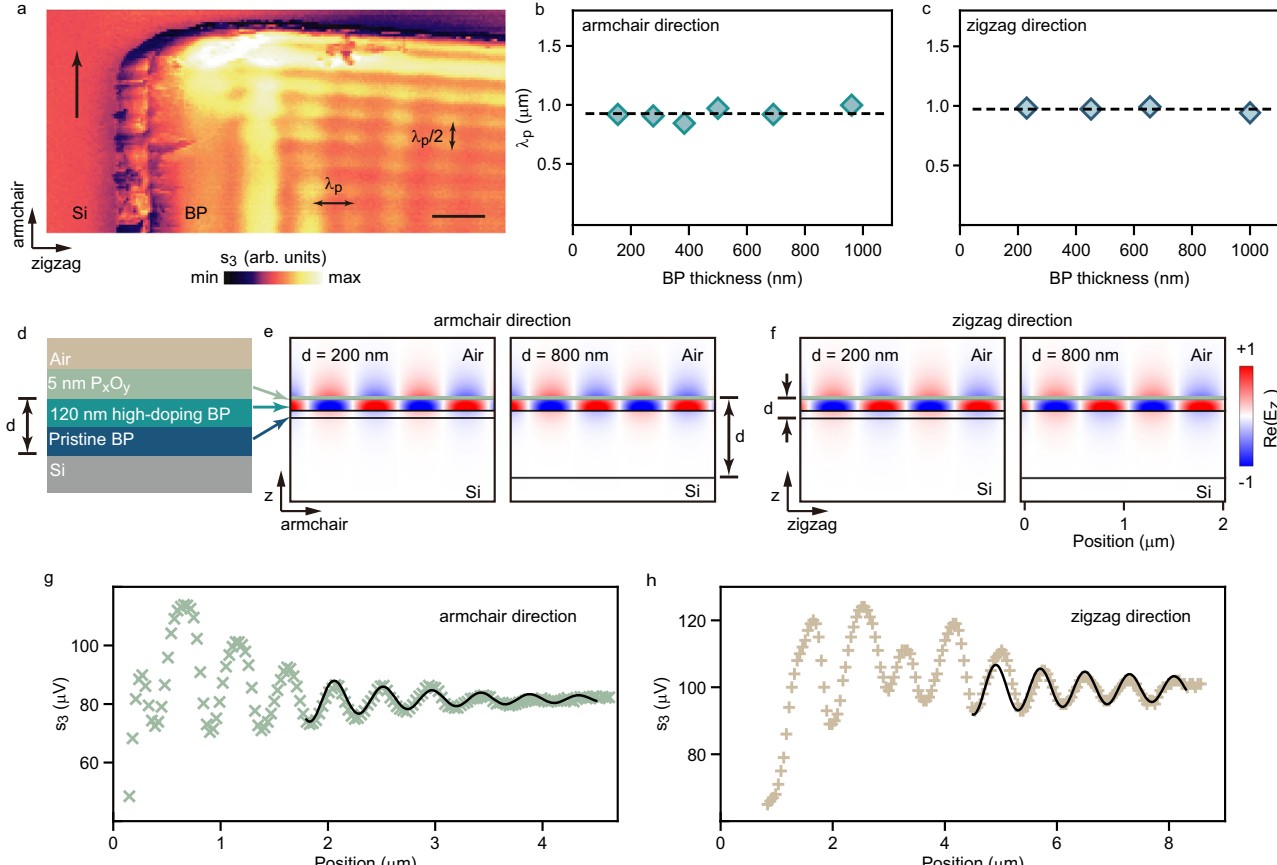

**Fig. 3 | Hyperbolic plasmons along the armchair and zigzag direction. a** Near-field amplitude images of polariton propagating along the armchair and the zigzag direction. The BP thickness is 700 nm, the scale bar is 1 μm, and the black arrow indicates the incident probing beam direction. **b, c** Plasmon wavelengths of the BP slabs with different thicknesses. **d** Schematic of the air/5-nm-thick $P_xO_y$/high-doping BP/pristine BP/Si five-layer stacking structure. The double arrow marked $d$ is the thickness of the BP. **e, f** Cross-sectional of the real part of z component electric field $Re(E_z)$ at $v/c = 950$ cm$^{-1}$ of the plasmon mode propagating along the armchair (**e**) and the zigzag (**f**) direction with the BP thickness $d = 200$ nm and 800 nm, respectively. **g, h** Low-loss plasmon propagating along the armchair (**g**) and the zigzag (**h**) direction. The symbols are experimental data. The solid lines are the fitting result of Eq. (4).

route, and the edge-launching + secondary-tip-launching + edge-emission route, respectively, $x$ is the tip-edge separation distance, $q_p$ is the momentum of plasmon, and $\varphi$ is a phase shift of each propagation route.

Equation (4) shows that at $\theta = 0°$ and 180°, only $A_1$ exists, leading to the long-period fringes govern; at $\theta = 90°$ and 270°, the destructive interference between $A_1$ and $A_2|\sin\theta|$ lessens the strength of the long-period fringe, leading to the short-period fringes $A_3|\sin\theta|$ domination (Cases for illumination from $\theta = 0°$ to 328° in Supplementary Note 3). The necessity to include the $A_3|\sin\theta|$ term in Eq. (4) implies that the BP edge-launched polaritons could further polarize the metal tip of the nanoscopy and induce a secondary tip-launched polariton, which is uncommon in van der Waals materials, such as the graphene, the h-BN, and the α-MoO₃[46,47,49]. The high efficiency of edge-launched polaritons underscores the role of the BP edge as an efficient optical antenna capable of converting free-space light into polaritons.

Figure 3a depicts the propagation of transient plasmons in a 700-nm-thick BP when the incident probe beam is perpendicular to the zigzag edge. As indicated in Fig. 2 previously, the fringe period in Fig. 3a equals $\lambda_p/2$ or $\lambda_p$ for the transient plasmons propagating along the armchair or zigzag direction, respectively. This results from the destructive interference between $A_1$ and $A_2|\sin 90°|$ for transient plasmons propagating along the armchair direction and the vanishing of $A_2|\sin 0°|$ and $A_3|\sin 0°|$ for transient plasmons propagating along the zigzag direction.

Intriguingly, the fringe period along the armchair direction of the 700-nm-thick BP in Fig. 3a is identical to that of the 353-nm-thick BP in Fig. 2b, which contradicts the volume polaritonic dispersion relation observed in other layered materials, where the polariton's wavelength increases with the slab's thickness[4,6,7]. According to the Beer–Lambert law, the bulky BP's pump absorption coefficient for $h\nu = 0.8$ eV photon is $\alpha_{pump} \sim 0.01$ nm$^{-1}$, implying that most photo-induced carriers exist in a space charge layer with a fixed thickness of about $\frac{1}{\alpha_{pump}}$[50–52]. For most polaritonic materials, there is no need to consider light penetration depth's influence on polaritonic properties. However, for transient plasmons, it is necessary to consider the impact of the pumping light's penetration depth because photo-induced carriers only exist where the pumping light can reach inside a material.

The BP slabs of 160–1000 nm thicknesses were used to confirm the space charge layer. Figure 3b, c shows the in-plane anisotropic transient plasmons wavelengths along the armchair direction $\lambda_{p, armchair} = 925$ nm and zigzag direction $\lambda_{p, zigzag} = 976$ nm are independent of the BP slab thickness, suggesting that transient plasmons exist solely within the space charge layer. Accordingly, the transient plasmons' wavelength is determined by the thickness of the space charge layer instead of the entire thickness of BP slabs.

Numerical simulations were conducted to gain further insight into the real part of z component electric field $Re(E_z)$ of the simplified 5-layer model in Fig. 3d. The BP slabs were exposed to ambient conditions in the experiment and suffered degradation[53]. Nevertheless, the

degradation did not deteriorate deeper into the BP due to a dense $P_xO_y$ layer of about 5-nm-thick on the surface[54–57], which has been considered in the simulation. The thickness of the space charge layer was optimized for 120 nm, equivalent to the reciprocal value of the pumping photons' absorption coefficient $\alpha_{pump}$. The simulated results in Fig. 3e, f show the independence of thickness of the BP ($d = 200$ nm and 800 nm) on the wavelength of the transient plasmons (at $v/c = 950$ cm$^{-1}$), which corresponds well with the experimental observation in Fig. 3b, c with $\lambda_{p, \text{armchair}} = 915$ nm and $\lambda_{p, \text{zigzag}} = 963$ nm.

Moreover, the superior carrier mobility of the BP allows long-range transient plasmon propagation[39]. Figure 3g, h shows the low-loss transient plasmons, with up to 9 bright stripes observed,

corresponding to a propagation length of approximately 10 μm. The curves were fitted using Eq. (4), leading to high-quality factors $Q = \frac{\text{Re}[q_p]}{\text{Im}[q_p]}$ with $Q_{\text{armchair}} = 22.18$ and $Q_{\text{zigzag}} = 18.59$. These $Q$ factors are comparable to those found in high-$Q$ plasmonic systems such as carbon nanotubes and suspended graphene[58,59], which is critical for constructing plasmonic connecting networks. Subsequently, the $Q$ factors were used to assess the long propagating decay time $\tau_p = \frac{Q}{v}$ with $\tau_{p, \text{armchair}} = 778$ fs and $\tau_{p, \text{zigzag}} = 652$ fs ($v/c = 950$ cm$^{-1}$ is the central frequency of hyperbolic plasmon in the Supplementary Note 5). The values of $\tau_{p, \text{armchair}}$, and $\tau_{p, \text{zigzag}}$ are consistent with the hot carrier scattering rates along the zigzag and armchair directions[60], indicating low propagating loss for the transient plasmons.

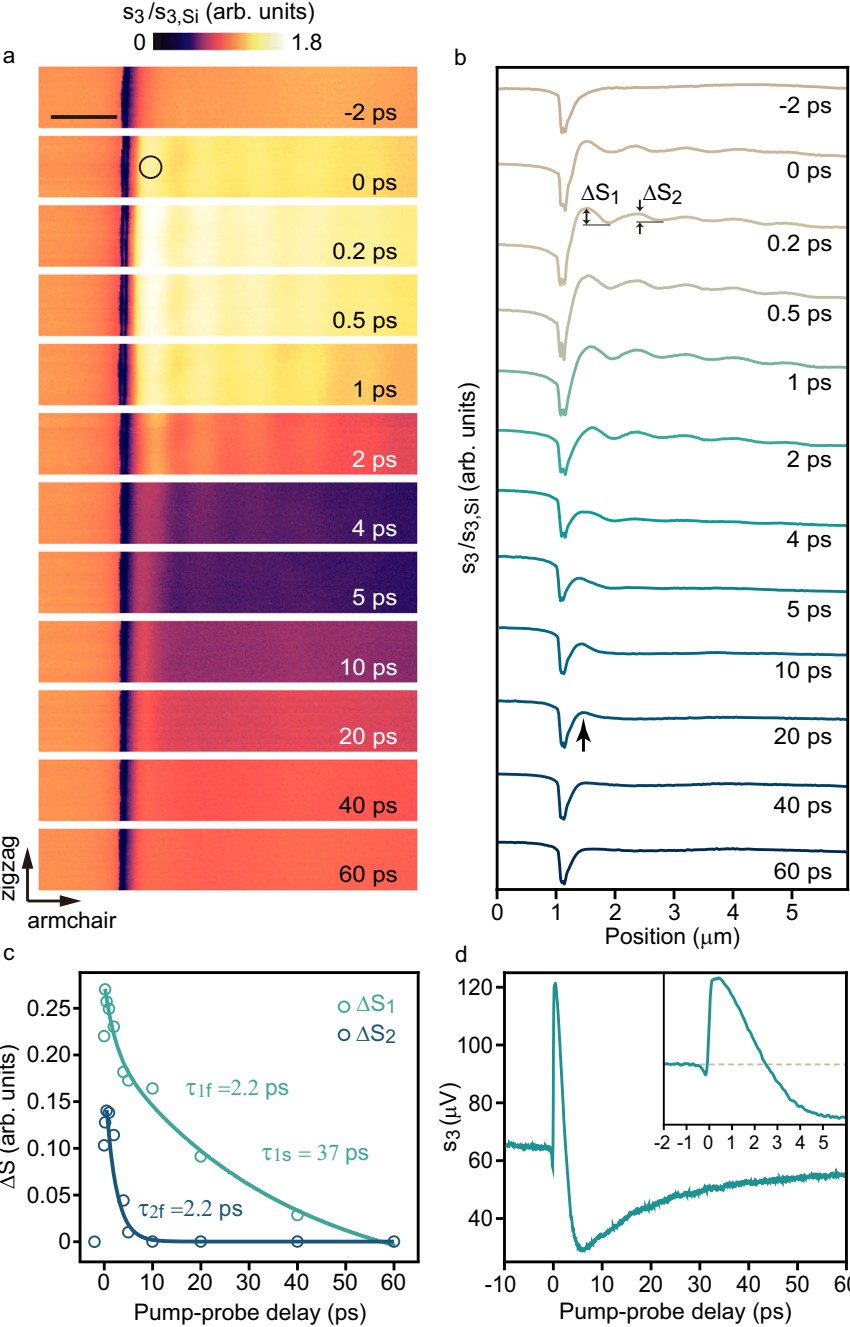

**Fig. 4 | Dynamic analysis of the transient plasmons. a** Normalized near-field amplitude $s_3/s_{3,Si}$ of a 280-nm-thick BP slab for twelve delay times $\tau$. Scale bar, 1 μm. **b** Near-field amplitude curves for the corresponding twelve different delay times $\tau$ in **a. c** Dynamics of the relative near-field intensity of the first ($\Delta S_1$) and the second bright

strip ($\Delta S_2$) in **b**. Opened circles are the experimental data, and solid lines are bi-exponential fitting for $\Delta S_1$ and exponential fitting for $\Delta S_2$, respectively. **d** Dynamics of the near-field amplitude $s_3$ from the black circle in **a**. The inset displays the $s_3$ at $\tau = -2$ to 6 ps, and the dashed line marks the $s_3$ level of the pristine state.

## Dynamics of transient plasmons

The BP transient hyperbolic plasmons are modes coupled by the probing photons and the non-equilibrium oscillating charge carriers. The propagation properties of the transient plasmons along the armchair direction are further analyzed, presented in Fig. 4, showing similarity to the results along the zigzag direction in supplementary Note 7. The normalized time-resolved near-field images $s_3/s_{3,Si}$ were captured at different time delays $\tau$ from −2 ps to 60 ps in Fig. 4a. At $\tau = -2$ ps, the absence of fringe is attributed to low initial carrier density. The increment in photo-induced carrier density at $\tau = 0$ to 0.2 ps results in clear fringes in the near-field images.

To further understand the dynamics of the transient plasmons, we extracted the relative strengths of the first ($\Delta S_1$) and the second ($\Delta S_2$) bright stripes from Fig. 4b. $\Delta S_1$ represents the impact of both the propagating transient plasmons and edge-related effects, while $\Delta S_2$ is considered purely related to the propagating transient plasmons. $\Delta S_1$ and $\Delta S_2$ show a bi-exponential and exponential decay in Fig. 4c, respectively, with a fast time constant of $\tau_{1f} = \tau_{2f} = 2.2$ ps, indicating the rapid reduction of the hot carrier density due to carrier trapping of defect states introduced by surface degradation[61]. However, a slow relaxation of $\Delta S_1$ with a time constant of $\tau_{1s} = 37$ ps is also present, showing a residual near-field intensity of $\Delta S_1 = 0.16$ at $\tau = 10$ ps due to the conductivity change and the carrier accumulation formed as an edge mode[62–65].

The time-resolved near-field images analysis reveals the presence of two transient plasmons: the ~5 ps propagating mode characterized by a series of fringes parallel to the edge and the ~40 ps localized edge mode embedded in the first bright fringe. Figure 4a and the corresponding line cuts in Fig. 4b show that the fringe contrast of the propagating transient plasmons depends on $\tau$, as the reduction of carrier density redshifts the frequency region of the propagating mode out of the fixed imaging bandwidth. Despite the decrease in carrier density over time, the spacing of the fringes remains nearly constant, as the broadband probing light covers the whole frequency span of the transient plasmons before their near-field amplitude decreases (Supplementary Note 5, 6)[34].

## Discussion

This letter introduced a promising approach to optically manipulate robust transient hyperbolic plasmons in the layered semiconductor black phosphorus using a dedicated ultrafast nanoscopy scheme. Optical pumping allows the BP's IFCs to topologically transit from the pristine ellipsoid to the non-equilibrium hyperboloid, exhibiting exotic non-equilibrium hyperbolic plasmon properties, such as the optically tunable plasmonic dispersion and the coexistence of different transient plasmonic modes. Therefore, layered semiconductors like black phosphorus may facilitate versatile optoelectronics applications with active dynamic optical control.

## Methods

### Preparation of the black phosphorus

The BP slabs were isolated from their crystal counterpart (Shanghai Onway Technology Co., Ltd) by mechanical exfoliation. By keeping the direction of each exfoliation consistent, the unique rectangular shape of the BP slabs could be realized. Then, the BP slabs were transferred onto the freshly cleaned Si substrates in a glove box (APURIS IGBS1800). Shallow $P_xO_y$ layers were formed during the exfoliation process for less than half an hour, and the samples were kept in the glove box except for the measurement.

### Ultrafast nanoscopy setup

The ultrafast nanoscopy contains three Erbium-doped fiber amplifiers connected to the same oscillator with a 76-fs pulse duration and 80 MHz repetition frequency (TOPTICA Photonics AG). Amplifiers 1 and 2 emitted near-infrared (NIR) pulses with a wavelength of 1500–1600 nm and a power of 400 mW, and amplifier 3 emitted supercontinuum pulses (980–2200 nm) using a nonlinear fiber. The pump branch was emitted from amplifier 1, and the probe branch was produced by difference frequency generation processes between amplifiers 2 and 3. By tuning the pitch angle of the nonlinear crystal (GaSe-1000H1, EKSMA OPTICS) to meet the phase-matching condition, The mid-infrared (MIR) broadband probe pulses ($v/c = 850{-}1200$ cm$^{-1}$) were obtained. The pump and probe pulses were spatially overlapped on the metalized Pt/Ir tip (ARROW-NCPt, Nanoworld) through a parabolic mirror of a commercial scattering-type scanning near-field optical microscopy (attocube systems AG). Samples were settled on a customized rotation platform to change the probe incident angles.

### Numerical simulations

The numerical simulation of the propagating of hyperbolic plasmon was calculated by the finite-element numerical simulation software (COMSOL). The boundary mode analysis and frequency domain module of COMSOL was used to simulate the near-field waveguide mode and propagation rays of volume hyperbolic plasmon, respectively.

## Data availability

Relevant data supporting the key findings of this study are available within the article and the Supplementary Information file. All raw data generated during the current study are available from the corresponding authors upon request.

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

## Acknowledgements

This work was supported by the National Key Research and Development Program of China (Grant Nos. 2022YFA1203500, J.C.), the National Natural Science Foundation of China (Grant Nos. 12204125, M.X.; 51971070, Q.L.; 10974037, Q.L.), the Strategic Priority Research Program of Chinese Academy of Sciences (Grant Nos. XDB 30000000, J.C.; XDB 36000000, Q.D.; XDA 09020300, Q.L.), Eu-FP7 Project (No. 247644, Q.L.). J.C. acknowledges the support from CAS Youth Interdisciplinary Team.

## Author contributions

R.F., Y.Q., and M.X. contributed equally to this work. M.X. and J.C. conceived the study. M.X. and R.F. performed the optical nanoscopy and Raman measurements. Y.Q., S.C., Q.L., and Q.D. fabricated the samples

and performed the TEM measurements. Y.Z. fabricated the gold disk. M.X., R.F., X.L., Y.Z., R.C., B.L., and H.W. performed the theoretic analysis and numerical simulations. J.C. supervised the work. M.X., R.F., and J.C. wrote the manuscript. All authors discussed the results and commented on the paper.

## Competing interests

The authors declare no competing interests.
