## [Peer Review File · Nature Communications]

Manipulating Hyperbolic Transient Plasmons in a Layered SemiconductorEditorial Note: Parts of this Peer Review File have been redacted as indicated to remove third-party material where no permission to publish could be obtained.

REVIEWER COMMENTS

Reviewer #1 (Remarks to the Author):

Questions the authors need to address:

-it is not clear how the authors get the dielectric function results shown in Fig. 1D. If the real of the dielectric function is negative, there is no need for pump-probe, BP's plasmons could be imaged with CW lasers. The authors need to clarify how these results were achieved, and whether they are supported by their or others work performed in determining the dielectric function, for example using ellipsometry or other methods.

-the work does not make it clear what the role of thickness is in the measured fringes, the authors need to clarify why the thicknesses used were selected and how thickness is related to fringe spacing?

-it is also not clear why the pump frequency were selected? The authors need to say clearly why the reasoning for the choice of the pump wavelength/ For example would a pump frequency of 532 nm give the same result?

-a related issue is will the authors see plasmonic modes they observe if they were to use CW laser rather than pump-probe? A few sentences justifying pump-probe requirement would be helpful to less experts along with the choice a specific pump laser frequency determination

-how was the anisotropy axis determined in BP? authors need to describe the process

-the authors need to address the effect of degradation on their measurements. BP degrades soon after exposure to ambient condition, even though it is exfoliated in ambient. Light (IR beam) also accelerates the degradation. How does degradation affect measurements?

-if the BP was encapsulated (for example by hBN), would the result be any different?

Suggestions:

-In the Abstract and elsewhere in the ms the authors use statements like: "In this letter, we

succeeded an active topological plasmonic dispersion transition in the black phosphorus 2D semiconductor with an optical carrier incubation, i...” since the BP studied is several layers thick, perhaps best to say “layered BP” instead of “2D BP”

-last paragraph of Introduction;

The photo-pumping process in work injects ample hot charge carriers into the BP, thereby triggering hyperbolic add... in our work

-Results and Discussion

In Fig. 1a, an experimental setup of the nanoscopy with...

add this

(see Methods)

Reviewer #2 (Remarks to the Author):

This manuscript explores the active topological plasmonic dispersion transition in black phosphorus (BP) induced by ultrafast laser pulses. This phenomenon can be understood through photo-induced electronic transitions, which result in changes in the dielectric tensors of BP. Utilizing ultrafast nanoscopy, the manuscript demonstrates the remarkable characteristics of these transient hyperbolic plasmons. While the transient polariton mode of the SiO₂/BP/SiO₂ heterostructure was previously demonstrated in Nature Nanotech 12, 207–211 (2017), that work did not achieve the active topological transition which is vitally important for optoelectronic applications. Furthermore, when compared to the study in Science 371, 617–620 (2021) where the photo-induced topological transition of WSe₂ was presented, the transient hyperbolic plasmons in BP discussed in this manuscript exhibit superior plasmonic properties, including a longer propagation length, efficient edge-excited plasmons, and coexistence of the short lifetime propagating mode and the long lifetime localized mode. Overall, this work indeed represents a significant fundamental improvement over prior research. Therefore, I would like to recommend this publication in Nature Communications after thoroughly addressing the following questions and comments.

1. In the first paragraph on page 4, the authors should provide a clearer explanation of how dielectric screening impacts the dielectric elements or provide a relevant citation.
2. In order to enhance the depiction of the photo-induced hyperbolic plasmonic response, it is suggested that the authors consider incorporating the near-field image s3 of a pristine BP/gold stacked structure in Fig. 1h.
3. In the first paragraph on page 6, it would be beneficial if the authors could provide further elucidation on why the influence of light penetration depth is not typically taken into account in most polaritonic

materials. Alternatively, they could support their statement by referencing a relevant citation that supports this assertion.

4. In the inset of Fig. 2d, the depiction of BP atop Si is misleading, potentially causing readers to interpret it as a cross-section of the sample. To address this concern, I suggest the authors to align the arrangement of the BP and Si slab in the inset with the configuration depicted in Fig. 2a.

5. The authors presented an interesting finding regarding the independence of BP thickness on the wavelength of transient plasmons. They attribute this phenomenon to the unchanging thickness (120 nm) of the space charge layer. Fig. S4 shows a clear trend where the polariton wavelength increases with the increasing thickness of doped BP. It is worth noting that this paper solely presents experimental results from BP samples with thicknesses exceeding 120 nm. To further strengthen the validity of the theoretical calculations depicted in Fig. S4, it is advisable to include measurements of the polariton wavelength for BP samples with thicknesses below 120 nm. This additional data would provide further support for the theoretical framework.

6. Fig. 4d shows the dynamics of the near-field amplitude intensity s_3 of the first bright fringe. Could the authors explain the significant decrease in intensity s_3 , which falls below the pristine level after 17 ps? Additionally, I am curious to know if this mechanism would have any effect on the fittings presented in Fig. 4c.

Reviewer #3 (Remarks to the Author):

In the manuscript, the authors reported pump-induced hyperbolic plasmons in black phosphorus (BP) visualized with near-field optical imaging and spectroscopy. The transient hyperbolic regime was first demonstrated by imaging the conical propagation of polaritons of BP placed on top of a Au disk launcher. Then the authors carried out systematic angle-dependent and thickness dependent measurements to further study the observed hyperbolic plasmon polaritons. I only have a few questions and comments before recommending for publication.

1. In Fig. 1 the authors demonstrated that transient hyperbolicity using nano-imaging. However, the probe frequency used for the imaging in Fig. 1h (850 -1200 cm^{-1}) seems to cover both hyperbolic and non-hyperbolic frequency range, according to Fig. 1d. It would be ideal if the same imaging experiment can be done using narrow-band probe, centering around e.g. 950 cm^{-1} (hyperbolic) and 1150 cm^{-1} (non-hyperbolic) to further demonstrate the crossover. The authors should also consider providing the topography image corresponding to Fig. 1h as well as the near-field imaging at negative time delay, similar to the ones in Fig. 4a.

2. What is the carrier density corresponding to 0.5 mJ/cm^2 pump fluence?

3. The nearly constant plasmon wavelength with varying BP thickness shown in Fig. 3 is interesting but also raises some question. I understand the argument about the finite penetration depth causing BP thicker than 100 nm to have the same transient plasmon wavelength. However, one can study BP films thinner than 100 nm, which should have homogeneous doping and will have more pronounced thickness dependence at e.g. 20 nm and 60 nm. Is the reason for lack of thinner BP samples in the current manuscript related to the surface degradation?

Minor issue, at page 7, second to last paragraph, the authors wrote “The low initial carrier density is attribute to the absence of polaritons fringes at $\tau=2$ ps.” I think the authors meant the other way around, i.e. absence of fringe is attributed to low carrier density.

Response Letter to Reviewers

Reviewer #1:

Comment 1-1: it is not clear how the authors get the dielectric function results shown in Fig. 1D. If the real of the dielectric function is negative, there is no need for pump-probe, BP's plasmons could be imaged with CW lasers. The authors need to clarify how these results were achieved, and what they are supported by theirs or others work performed in determining the dielectric function, for example using ellipsometry or other methods.

Response 1-1: The authors thank the reviewer for this comment.

As shown in Fig.1c, the dipole model was used to fit the near-field spectra, and the dielectric elements of BP are the fitting parameters. Thus, by substituting the effective dielectric constant $\langle \epsilon \rangle = \sqrt{\langle \epsilon_{plane} \rangle \epsilon_z}$ into the dipole model to fit the spectra in Fig.1c, we obtained the dielectric elements of BP in Fig.1d.

The dielectric elements of the pristine BP are $\epsilon_{armchair} = 18$, $\epsilon_{zigzag} = 14$, and $\epsilon_z = 9.75$ (the dotted lines in Fig.1d). The pristine BP's positive dielectric values do not support hyperbolic plasmon. Therefore, optical pumping is needed to excite photo-induced carriers to turn ϵ_z to negative.

The dielectric values extracted from the near-field spectra agree with ellipsometry in general. Several works have verified near-field spectra to be a high-accuracy method. (*Physical Review B* 2014, 90(8), 085136; *ACS Photonics* 2021, 8, 1, 175–181; *ACS Nano* 2014 8, 7, 6911; *Applied Surface Science* 2022, 574, 151672).

An additional remark is that the dielectric values should reflect the non-equilibrium state BP to calculate the transient plasmons. Therefore, usual ellipsometry does not apply to measure dielectric value in a non-equilibrium state directly.

Comment 1-2: the work does not make it clear what the role of thickness is in the measured fringes, the authors need to clarify why the thicknesses used were selected and how thickness is related to fringe spacing?

Response 1-2: The authors thank the reviewer for this comment.

We measured many 160 – 1000-nm-thick BPs and realized that the fringe spacing is consistent in these different thick BPs. This finding is attributed to BP's 120-nm-thick surface charge layer. The simulations in Fig. 3e–f show that the optical fields of the hyperbolic plasmons are confined in this 120-nm-thick surface charge layer, and the wavelengths of plasmons of 200 and 800-nm-thick BP are the same. These thickness-independent transient plasmons in BP are very different from other equilibrium polaritons, such as phonon polaritons.

We are studying the non-equilibrium plasmons in BP, including the dispersion, the edge effects, the incident light dependence, and the dynamics; thus, stable samples are essential. We prefer thicker BPs to conduct measurements because they are relatively robust and easy to exfoliate. The BPs in the main manuscript figures are all hundreds of nanometers thick.

Comment 1-3: it is also not clear why the pump frequency were selected? The authors need to say clearly why the reasoning for the choice of the pump wavelength/ For example would a pump frequency of 532 nm give the same result?

Response 1-3: The authors thank the reviewer for this comment.

As introduced in the method part, our laser setup is based on three erbium-doped fiber lasers. The output photon energy of this laser is 0.8 eV, which is larger than the 0.3 eV band gap of bulk BP and able to generate electronic transitions from the valence band to the conduction band in BP.

In principle, if the laser photon energy is larger than the bandgap of a semiconductor, it is sufficient to excite electrons to the conduction band. However, when it refers to a particular material, the pumping efficiency relates to more factors.

When referring to employing a 532 nm pumping laser, we believe the BP can be excited to a non-equilibrium state. However, since the anisotropic dielectric function of BP, the dielectric element ϵ_i ($i = x, y, z$) along three different directions can be either all negative, two of them are negative, or just one direction is negative, which could introduce versatile iso-frequency contours (IFCs) possibilities, and needs to be verified by experiment. Therefore, we would anticipate that using 532 nm pumping in principle

might present several possible outcomes, including the same result in this work.

Comment 1-4: a related issue is will the authors see plasmonic modes they observe if they were to use CW laser rather than pump-probe? A few sentences justifying pump-probe requirement would be helpful to less experts along with the choice a specific pump laser frequency determination.

Response 1-4: The authors thank the reviewer for this comment. We performed the experiments with monochromatic near-field imaging and added this content to Fig. S12.

Due to the low initial carrier density, the pristine BP does not support hyperbolic plasmons. We performed monochromatic near-field imaging on a 205-nm-thick BP slab using a quantum cascade laser (DRS Daylight MIRcat). The tuning range of the laser frequency covers 940 – 1160 cm^{-1} . Fig. S12b-i shows the monochromatic near-field imaging under different frequencies, where the BP exhibits uniform near-field signal from 940 – 1160 cm^{-1} . Fig. S12j shows the near-field profiles from the white dashed line in Fig. S12b. The near-field profiles do not show plasmonic fringe from 940 – 1160 cm^{-1} . Another approach to probing BP plasmons is to measure with a THz laser that might match the pristine BP's low doping conditions.

A pump-probe investigation scheme is necessary to generate sufficient carriers in BP and explore its plasmonic properties. In principle, a pumping light with photon energy higher than the bandgap is adequate to generate free carriers (Fig. 1b). Since we do not have many different lasers to test the best pumping conditions comprehensively, we can only summarize our limited experience from the InSb study (*Adv. Mater.* 2023, 35, 2208952). For the plasmon frequency $\omega_p \propto \sqrt{n}$, the tunable frequency range of ω_p is determined by the maximum n , which depends on the density state of the electron at the conduction band. Hence, in principle, the saturation electron concentration and the up-limit of plasmon frequency can be elevated through high photon energy pumping excitation.

Fig. S12. Monochromatic near-field amplitude images of the pristine BP. a) The topographic of 205 nm-thick BP. Scale bar: 1 μm . b) – i) Monochromatic near-field amplitude images of BP from 940 – 1160 cm^{-1} . j) Near-field profiles from the position marked by the white dashed line in (b).

Comment 1-5: how was the anisotropy axis determined in BP? authors need to describe the process.

Response 1-5: The authors thank the reviewer for this comment.

In the supplementary material Fig. S1 and "the TEM and Raman spectra" part, we introduced how to use TEM and Raman spectroscopy to characterize the anisotropy axis.

In Raman spectroscopy, we recorded the intensity of A_g^2 mode at different θ ; θ is the angle between the long straight edge of BP and the laser polarization. When the polarization is perpendicular to the zigzag direction of the BP ($\theta = 90^\circ$ and 270°), the A_g^2 intensity becomes the strongest. Therefore, by using the polarized Raman shown in Fig. S1f, we could confirm that the long straight edge of BP is parallel to the zigzag direction.

We also carried out TEM characterization. By comparing the atomic arrangement of BP from the TEM image (Fig. S1d) and the atomic structure of BP (Fig. S1b), we verified that the long straight edge of BP is parallel to the zigzag direction.

As references, BP ribbons through mechanical exfoliation as ours have also been investigated by many others from different perspectives, and the results have

consistently shown that the long straight edge of BP is parallel to zigzag (*Small* 2023, 19, 2207538; *J. Raman Spectrosc.* 2018, 49, 76; *AIP Advances* 2018, 8, 105216; *Nanoscale* 2015, 7, 18708).

Fig. S1. TEM and Raman spectra of rectangular BP ribbons. **a)** The optical image of rectangular shape BP slabs. **b)** The atomic configuration of the BP. **c)** The transmission electron microscopy (TEM), and **d)** the high-resolution TEM image taken from the red rectangle region in **(c)**. **e)** Raman spectrum of the BP slabs. The wavelength of the excitation laser is 532 nm. The excitation and collected light polarization are fixed to be parallel. **f)** The angular-dependent intensity for the A_g^2 mode. The angle in the polar coordinate is the angle between the laser polarization direction and the long straight edge of the BP.

Comment 1-6: the authors need to address the effect of degradation on their measurements. BP degrades soon after exposure to ambient condition, even though it is exfoliated in ambient. Light (IR beam) also accelerates the degradation. How does degradation affect measurements?

Response 1-6: The authors thank the reviewer for this comment. We conducted the experiments on degraded BP and added the result to supplementary material Fig. S11.

We performed near-field imaging on the same BP with different degradation levels. Fig. S11a-b was captured from freshly exfoliated BP. Fig. S11c-d was acquired from BP that had been exposed to atmospheric conditions for three days. Upon close examination of the topography in Fig. S11c, we observe the presence of point-like protrusions. This characteristic indicates the typical degradation of BP.

The dark spots in Fig. S11d also reveal the evident impact of this degradation. It's worth noting that the BP's hyperbolic plasmon is a bulk mode, similar to the phonon polariton in hBN (*Science* 2014, 343, 1125; *Nat Commun.* 2015, 6, 6963). Therefore, the electric field of plasmons is primarily confined inside the BP (Fig. 3e-f), which means that the presence of a surface degradation layer, although visually apparent, we can still observe the faint BP's plasmonic modes in Fig. S11d.

Fig. S11. Effects of BP's degradation on hyperbolic plasmons. a) – b) The topography (a) and the near-field amplitude image (b) of the freshly exfoliated BP. Scale bar: 1 μm . The frequency region of the probe is $\nu/c = 850 - 1200 \text{ cm}^{-1}$. c) – d) The

topography (c) and the near-field amplitude image (d) of BP after being exposed to atmospheric conditions for 3 days. The inset in a) and c) shows the topography detail of the black square with a 200 nm scale bar in red.

Comment 1-7: if the BP was encapsulated (for example by hBN), would the result be any different?

Response 1-7: The authors thank the reviewer for this comment. We made new samples for experiments and added the content to supplementary material Fig. S10.

In Fig. S10a, we covered BP with a 10-nm-thick hBN. The spectral dip observed near 800 cm^{-1} in Fig. S10c corresponds to the absorption of hBN (*Nanoscale Advances* 2019, 1, 1693-1701). In Fig. S10b, we also observed BP's photo-induced plasmon fringes. This photo-induced plasmon fringe spacing is consistent with the results obtained from bare BP in the main text.

In principle, hBN could protect BP from degradation. However, in Fig. S1a, the size of the BP is on the order of hundreds of microns. To ensure effective near-field detection of BP's plasmon by the s-SNOM tip, we couldn't use a thick hBN protection layer. Our mechanically exfoliated 10-nm-thick hBN layer could only cover a small portion of the BP, leaving most of the BP exposed to the atmosphere and susceptible to degradation. Unfortunately, the degradation could extend to the regions covered by hBN, rendering hBN's protective function ineffective.

Fig. S10. Hyperbolic plasmons in BP covered with a 10-nm-thick hBN protecting layer. a) The topography image of the hBN-protected BP. Scale bar: $1\text{ }\mu\text{m}$. b) The Near-field amplitude image of hBN-protected BP. The frequency region of the probe is $\nu/c = 850 - 1200\text{ cm}^{-1}$. c) Nano-FTIR of hBN protected BP from the black circle in (b).

Comment 1-8: In the Abstract and elsewhere in the ms the authors use statements like: "In this letter, we succeeded an active topological plasmonic dispersion transition in the black phosphorus 2D semiconductor with an optical carrier incubation, i..." since the BP studied is several layers thick, perhaps best to say "layered BP" instead of "2D BP"

Response 1-8: We thank the reviewer for these suggestions and corrections. We have substituted the "2D BP" for "*layered BP*" in the revised manuscript.

Comment 1-9: last paragraph of Introduction, "The photo-pumping process in work injects ample hot charge carriers into the BP, thereby triggering hyperbolic". add... in our work.

Response 1-9: We thank the reviewer for these suggestions and corrections and have made corresponding revisions to the manuscript. On **line 2, page 3**, we have replaced the initial content from "The photo-pumping process injects ample hot charge carriers into the BP, thereby triggering hyperbolic plasmons that are otherwise absent in the pristine state" to "*The photo-pumping process in our work injects ample hot charge carriers into the BP, thereby triggering hyperbolic plasmons that are otherwise absent in the pristine state*".

Comment 1-10: Results and Discussion. "In Fig. 1a, an experimental setup of the nanoscopy with..." add this "see Methods".

Response 1-10: We thank the reviewer for these suggestions and corrections and have made corresponding revisions to the manuscript. On **line 12, page 3**, we have replaced the initial content from "In Fig. 1a, an experimental setup of the nanoscopy with ultrafast lasers was shown, which is utilized to study the photo-induced hyperbolic plasmonic response of the BP slabs" to "*In Fig. 1a, an experimental setup of the nanoscopy with ultrafast lasers was shown, which is utilized to study the photo-induced hyperbolic plasmonic response of the BP slabs (see Methods)*".

Reviewer #2:

General comment: This manuscript explores the active topological plasmonic dispersion transition in black phosphorus (BP) induced by ultrafast laser pulses. This phenomenon can be understood through photo-induced electronic transitions, which result in changes in the dielectric tensors of BP. Utilizing ultrafast nanoscopy, the manuscript demonstrates the remarkable characteristics of these transient hyperbolic plasmons. While the transient polariton mode of the SiO₂/BP/SiO₂ heterostructure was previously demonstrated in Nature Nanotech 12, 207–211 (2017), that work did not achieve the active topological transition which is vitally important for optoelectronic applications. Furthermore, when compared to the study in Science 371, 617–620 (2021) where the photo-induced topological transition of WSe₂ was presented, the transient hyperbolic plasmons in BP discussed in this manuscript exhibit superior plasmonic properties, including a longer propagation length, efficient edge-excited plasmons, and coexistence of the short lifetime propagating mode and the long lifetime localized mode. Overall, this work indeed represents a significant fundamental improvement over prior research. Therefore, I would like to recommend this publication in Nature Communications after thoroughly addressing the following questions and comments.

General response: The authors appreciate the reviewer's positive assessment of this work. As the reviewer mentioned, this work demonstrates unique hyperbolic polaritonic phenomena in BP, such as photo-induced topological transitions, long-range propagation, effective edge excitations, and polaritons with different relaxation times. The authors thank the reviewers for valuable suggestions to improve this work. We have provided detailed responses after each corresponding comment.

Comment 2-1: In the first paragraph on page 4, the authors should provide a clearer explanation of how dielectric screening impacts the dielectric elements or provide a relevant citation.

Response 2-1: We thank the referee for this comment.

Equation 1 $\epsilon_j = \epsilon_{\infty,j} - \epsilon_{pl,j} = \epsilon_{\infty,j} - \frac{\omega_{p,j}^2}{\omega^2 + i\omega\gamma_j}$; $j = x, y, z$ shows that the dielectric element's value ϵ_j is derived from subtracting the plasmon term $\epsilon_{pl,j}$ from the

high-frequency permittivity $\epsilon_{\infty,j}$. Equation 1 implies that a positive $\epsilon_{\infty,j}$ weakens the negative plasmon-induced change $\epsilon_{pl,j}$ in the ϵ_j , which is the dielectric screening effect mentioned in ref 37 (*Physical Review B*, 2017, 95, 245140). We have replaced the Equation 1 " $\epsilon_j = \epsilon_{\infty,j} - \frac{\omega_{p,j}^2}{\omega^2 + i\omega\gamma_j}$; $j = x, y, z$ " to " $\epsilon_j = \epsilon_{\infty,j} - \epsilon_{pl,j} = \epsilon_{\infty,j} - \frac{\omega_{p,j}^2}{\omega^2 + i\omega\gamma_j}$; $j = x, y, z$ " on **line 18, page 3** in the revised manuscript. We added the dielectric screening effect description: "*Equation (1) indicates that ϵ_j is obtained by subtracting the $\epsilon_{pl,j}$ from the $\epsilon_{\infty,j}$, which signifies that a positive $\epsilon_{\infty,j}$ weakens the negative change induced by $\epsilon_{pl,j}$ in ϵ_j , the dielectric screening effect.*" in the revised manuscript on **line 22, page 3**.

In the case of BP, the $\epsilon_{\infty,j}$ for the armchair, zigzag, and out-of-plane directions are 18, 14, and 9.75, respectively. Consequently, under optical pumping excitation, the ϵ_z for the out-of-plane direction becomes negative after deducting the plasmon term due to the smaller $\epsilon_{\infty,z} = 9.75$, while the ϵ_{ac} or ϵ_{zz} for the armchair and zigzag directions remain positive due to a much larger $\epsilon_{\infty,ac} = 18$ and $\epsilon_{\infty,zz} = 14$. The dielectric screenings in the armchair and zigzag directions are stronger than those in the out-of-plane direction.

Comment 2-2: In order to enhance the depiction of the photo-induced hyperbolic plasmonic response, it is suggested that the authors consider incorporating the near-field image s3 of a pristine BP/gold stacked structure in Fig. 1h.

Response 2-2: We thank the referee for bringing up the issue regarding Fig. 1h. Authors fabricated new nano gold disks and BP to verify the hyperbolic plasmon. The results are added to the revised Fig. 1h and new Fig. S8.

The near-field images of pumped and unpumped BP on the gold sample have been updated in Fig. 1h, and more detailed data of the gold disk have been updated in Fig. S8. In the updated Fig. 1h, the thicknesses of BP and gold disk are 310 nm and 100 nm, respectively. The topography image in Fig. S8a shows that, after BP coverage, the region where the gold disk below is slightly arched. The hyperbolic IFCs allow the gold

disk's edge to launch conical-shaped energy rays, producing two bright rings separated by a dark ring above the round gold disk's edge. Therefore, in Fig. S8b, the near-field amplitude image under 0.5 mJ/cm^2 pump illuminated clearly shows a dark ring sandwiched by two bright rings above the gold disk. In Fig. S8c, the pristine BP's near-field image does not show particular optical features only with probe light illumination. In Fig. S8d, we set $\tau = -2 \text{ ps}$ under 0.5 mJ/cm^2 pumping light so that the probe shines the BP before the pump comes. In this case, the probed BP is also pristine, and the near-field amplitude image neither exhibits polaritonic rings. Therefore, the pump-injected non-equilibrium carriers are crucial to launch BP's hyperbolic plasmons. In the revised manuscript, we have replaced the "Accordingly, the near-field image in Fig. 1h of the non-equilibrium BP/gold stacked structure clearly shows a dark ring sandwiched by two bright rings above the gold disk, confirming the photo-induced out-of-plane hyperbolic plasmons in BP" to "Accordingly, the near-field image in Fig. 1h of the non-equilibrium BP/gold stacked structure clearly shows a dark ring sandwiched by two bright rings above the gold disk, while the pristine BP/gold does not exhibit any ring pattern, confirming the excitation of the photo-induced out-of-plane hyperbolic transient plasmons in BP (Supplementary Fig. S8)" on line 2, page 5.

Updated Fig. 1

Original Fig. 1

Fig. S8. Photo-induced plasmonic ring of BP/gold disk stacked structure. a) The topography of BP/gold stacked structure. b) - d) Near-field amplitude imaging with 0.5 mJ/cm² pump fluence (b) and 0 mJ/cm² pump fluence (c) and 0.5 mJ/cm² pump fluence at -2 ps (d).

Comment 2-3: In the first paragraph on page 6, it would be beneficial if the authors could provide further elucidation on why the influence of light penetration depth is not typically taken into account in most polaritonic materials. Alternatively, they could support their statement by referencing a relevant citation that supports this assertion.

Response 2-3: The authors thank the referee for this comment.

The transient plasmons are very different from other polaritons. For example,

common plasmons and phonon polaritons are the electron and phonon collective movements coupled with probe light in an equilibrium state, meaning that the number of free carriers and phonons remain unchanged when the probe light interplays. However, there are not enough free carriers in pristine BP to collectively interact with the probe light to exhibit a plasmonic response; therefore, optical pumping is employed to generate free carriers in BP, both electrons and holes, to a non-equilibrium state to couple with probe light collectively exhibiting transient plasmonic response. Because these optically induced carriers are excited by pumping light inside the material, estimating the volume scale of a given material in which photon-induced carriers exist is necessary. The pumping light's frequency determines the optical penetration depth scale, which is why we need to consider light penetration depth in this case of optically pumped non-equilibrium state plasmons in BP. In the revised manuscript, we have added the description: "*For most polaritonic materials, there is no need to consider light penetration depth's influence on polaritonic properties. However, for transient plasmons, it is necessary to consider the impact of pumping light's penetration depth because photo-induced carriers only exist where the pumping light can reach inside a material*". On line 21, page 6.

Similar concepts of considering the volume scale of carriers also can be found. For example, in the study of terahertz plasmon on topological insulators Bi_2Se_3 (*Nat Commun.* 2022 13, 1374), Chen built various theoretical models of charge distributions in Bi_2Se_3 , including homogeneous distribution (uniform bulk doping) and inhomogeneous distribution (low bulk doping and surface electron gas), and demonstrated that these differing space charge distributions lead to distinct polariton dispersions. The agreement between the inhomogeneous model and the experimental results indicates that it is necessary to consider the volume scale of carriers in plasmonic analysis.

Comment 2-4: In the inset of Fig. 2d, the depiction of BP atop Si is misleading, potentially causing readers to interpret it as a cross-section of the sample. To address this concern, I suggest the authors to align the arrangement of the BP and Si slab in the

inset with the configuration depicted in Fig. 2a.

Response 2-4: The authors thank the referee for pointing out the deficiencies in Fig. 2. We have updated Fig. 2 as suggested by the referee.

The updated Fig. 2

The original Fig. 2

Comment 2-5: The authors presented an interesting finding regarding the independence of BP thickness on the wavelength of transient plasmons. They attribute this phenomenon to the unchanging thickness (120 nm) of the space charge layer. Fig. S4 shows a clear trend where the polariton wavelength increases with the increasing thickness of doped BP. It is worth noting that this paper solely presents experimental results from BP samples with thicknesses exceeding 120 nm. To further strengthen the validity of the theoretical calculations depicted in Fig. S4, it is advisable to include measurements of the polariton wavelength for BP samples with thicknesses below 120 nm. This additional data would provide further support for the theoretical framework.

Response 2-5: The authors thank the referee for this comment. We fabricated new samples, conducted relevant experiments, and added the content to Fig. S9.

In Fig. S9a-d, we selected four different BP slabs with thicknesses from 21 – 164 nm to explore the thickness-related polariton fringes. In Fig. S9e-h, we found that the photo-induced plasmon fringes are only apparent in the BP slabs thicker than 102 nm, and the fringe contrast increases significantly with the increase in BP's thickness.

As the thickness of BP decreases, the fringe spacing in Fig. S9g (102 nm thick BP) reduces by about 13% compared with Fig. S9h (equivalent to 120 nm thick BP), which shows the BP's hyperbolic transient plasmon fringe spacing almost linearly decreases with the reduction in BP thickness, similar to the case of hBN phonon polaritons (*Science* 2014, 343, 1125). The cause of fringe vanishing in thin BP flakes may be twofold. One is that the oxidization rate of thin BPs becomes higher (*Nat. Mater.* 2015, 14, 826–832), leading to an increasing plasmon propagation loss due to impurities scattering (Fig. S11; *Sci. Adv.* 2022, 8, eabn0627). The thinner the flake is, the stronger the oxidization effects. On the other hand, thin BP flakes exhibit a relatively weaker plasmonic response, leading to a decrease in the near-field amplitude signal of plasmon. Similar fringe contrast deduction with flake thickness also occurs in typical hyperbolic materials such as hBN and MoO₃, where the fringe contrast of polaritons decreases rapidly with decreasing thickness (*Adv. Mater.* 2018, 30, 1705318; *Adv. Mater.* 2019, 31, 1806603). Here, we cite the graph in *Adv. Mater.* 2018, 30, 1705318 to show similar results.

The thin BP samples degrade more rapidly in the atmospheric environment and do not last a long measurement time essential for dynamic near-field measurements. Therefore, we prefer thick samples exceeding 200 nm for studying the non-equilibrium plasmons in BP.

Fig. S9. Plasmons of different thin BP slabs. a) – d) Topography of BPs with a thickness of 21 – 164 nm. Scale bar: 1 μm . e) – h) Near-field amplitude images of BP with a thickness of 21 – 164 nm. The frequency region of the probe is $\nu/c = 850 - 1200 \text{ cm}^{-1}$.

[REDACTED]

Fig. R1. s-SNOM images of the $\alpha\text{-MoO}_3$ flakes with different thicknesses. It is taken from *Adv. Mater.* 2018, 30, 1705318.

Comment 2-6: Fig. 4d shows the dynamics of the near-field amplitude intensity s_3 of the first bright fringe. Could the authors explain the significant decrease in intensity s_3 , which falls below the pristine level after 17 ps? Additionally, I am curious to know if this mechanism would have any effect on the fittings presented in Fig. 4c.

Response 2-6: The authors thank the referee for raising this question.

We found an error in the original Fig. 4d. The x-axis is incorrectly labeled, where 1 ps was mismarked as 10 ps. Therefore, in original Fig. 4d, 17 ps should be 1.7 ps. We apologize for this error. We made new samples and reconducted measurements, extending the probe delay to 60 ps and plotting new figures in Fig. 4. The new and old dynamics data show consistent trends.

The downward signal after 1.7 ps can be understood in principle: The near-field

signal is not directly proportional to the charge carrier concentration but follows a trend determined by the Point Dipole / Finite Dipole Model. The near-field scattering amplitude s_3 is proportional to $|\sum_{j=1}^{\infty} a_j \beta^j|$, where a_j are the coefficients, $\beta = \frac{\varepsilon-1}{\varepsilon+1}$, and ε is the dielectric constant (*J. Phys. Chem. Lett.* 2013, 4, 1526). Interestingly, when $\varepsilon \sim -1$, the near-field reflection coefficient $\beta = \frac{\varepsilon-1}{\varepsilon+1}$ diverges, giving rise to large near-field amplitude signals and near-field reflectivity that can be larger than the pristine level. As the carrier density decreases, at some point, the permittivity will be $\varepsilon \sim +1$, and in this case, the near-field reflection is minimal: $\beta = \frac{\varepsilon-1}{\varepsilon+1} \rightarrow 0$. As ε becomes larger than 1, β increases. We cite a graph from *Nano Lett.* 2010, 10, 4, 1387 as Fig. R2 shows, the near-field signal increases and then drops to a minimum and increases from the free-carrier concentration $n = 10^{20}$ to 10^{18} cm^{-3} , similarly corresponding to the photo-induced carrier concentration decay dynamics.

The decrease of relative strengths ($\Delta S_{1,2}$) in Fig. 4c and the rapid decline of the near-field signal after 2.5 ps in Fig. 4d can generally be understood as related to transient plasmonic decay and ε variations, which are connected but different. Therefore, the fittings presented in Fig. 4c are not directly determined by the mechanism that near-field signal S_3 falls below the pristine level.

The updated Fig. 4

The original Fig. 4

[REDACTED]

Fig. R2. Near-field signal vs carrier concentration. It is taken from *Nano Lett.*

2010, 10, 4, 1387.

Reviewer #3:

General comment: In the manuscript, the authors reported pump-induced hyperbolic plasmons in black phosphorus (BP) visualized with near-field optical imaging and spectroscopy. The transient hyperbolic regime was first demonstrated by imaging the conical propagation of polaritons of BP placed on top of Au disk launcher. Then the authors carried out systematic angle-dependent and thickness dependent measurements to further study the observed hyperbolic plasmon polaritons. I only have a few questions and comments before recommending for publication.

General response: The authors very much appreciate the recommendation from the referee and have made changes carefully according to the referee's suggestions and comments.

Comment 3-1: In Fig. 1 the authors demonstrated that transient hyperbolicity using nano-imaging. However, the probe frequency used for the imaging in Fig. 1h (850 - 1200 cm^{-1}) seems to cover both hyperbolic and non-hyperbolic frequency range, according to Fig. 1d. It would be ideal if the same imaging experiment can be done using narrow-band probe, centering around e.g. 950 cm^{-1} (hyperbolic) and 1150 cm^{-1} (non-hyperbolic) to further demonstrate the crossover. The authors should also consider providing the topography image corresponding to Fig. 1h as well as the near-field imaging at negative time delay, similar to the ones in Fig. 4a.

Response 3-1: We thank the referee for this comment.

Since the relaxation time of BP's carriers is several picoseconds, we need to use probe light of 100 fs to carry out near-field imaging, which results in a broad spectral width of 350 cm^{-1} . For WSe_2 with 50 ps slow relaxation, the probe with narrow spectral ranges can be used to perform quasi-monochromatic near-field imaging. However, the probe's duration reaches 500 fs in that situation (*Science* 2021, 371,617). Thus, a narrow band probe with a 500 fs duration makes it challenging to track BP's transient plasmons with a fast decay of 2.2 ps in Fig. 4a-c.

We made new samples and reperformed measurements. The near-field images of pumped and unpumped BP on the gold sample have been updated in Fig. 1h, and more detailed data of the gold disk have been updated in Fig. S8. In the updated Fig. 1h, the thicknesses of BP and gold disk are 310 nm and 100 nm, respectively. The topography image in Fig. S8a shows that, after BP coverage, the region where the gold disk below is slightly arched. The hyperbolic IFCs allow the gold disk's edge to launch conical-shaped energy rays, producing two bright rings separated by a dark ring above the round gold disk's edge. Therefore, in Fig. S8b, the near-field amplitude image under 0.5 mJ/cm^2 pump illuminated clearly shows a dark ring sandwiched by two bright rings above the gold disk. In Fig. S8c, the pristine BP's near-field image does not show particular optical features only with probe light illumination. In Fig. S8d, we set $\tau = -2$ ps under 0.5 mJ/cm^2 pumping light so that the probe shines the BP before the pump comes. In this case, the probed BP is also pristine, and the near-field amplitude image neither exhibits polaritonic rings. Therefore, the pump-injected non-equilibrium carriers are crucial to launch BP's hyperbolic plasmons. In the revised manuscript, we have replaced the "Accordingly, the near-field image in Fig. 1h of the non-equilibrium BP/gold stacked structure clearly shows a dark ring sandwiched by two bright rings above the gold disk, confirming the photo-induced out-of-plane hyperbolic plasmons in BP" to "*Accordingly, the near-field image in Fig. 1h of the non-equilibrium BP/gold stacked structure clearly shows a dark ring sandwiched by two bright rings above the gold disk, while the pristine BP/gold does not exhibit any ring pattern, confirming the excitation of the photo-induced out-of-plane hyperbolic transient plasmons in BP (Supplementary Fig. S8)*" on line 2, page 5.

Updated Fig. 1

Original Fig. 1

Fig. S8. Photo-induced plasmonic ring of BP/gold disk stacked structure. a) The topography of BP/gold stacked structure. b) - d) Near-field amplitude imaging with 0.5

mJ/cm² pump fluence (b) and 0 mJ/cm² pump fluence (c) and 0.5 mJ/cm² pump fluence at -2 ps (d).

Comment 3-2: What is the carrier density corresponding to 0.5 mJ/cm² pump fluence?

Response 3-2: The authors thank the reviewer for this comment.

Through the formula: $m_{(e,h),j}^* = \hbar^2 k_j \frac{\partial k_j}{\partial E_{(C,V)}}$ (*Advances in Physics* 1974, 23, 435-522), we can get the effective mass of electrons and holes of BP in the non-equilibrium state of 0.8 eV pumping and then through the formula: $m_j^* = \frac{m_{e,j}^* m_{h,j}^*}{m_{e,j}^* + m_{h,j}^*}$, we obtained the reduced effective mass of bulk BP in different directions, where $m_x^* = 0.12m_0$, $m_y^* = 0.32m_0$, $m_z^* = 0.10m_0$, m_0 is the electron mass. By bringing the m_j^* into the $\omega_{p,j} = \sqrt{\frac{ne^2}{m_j^* \epsilon_0}}$, we got the carrier concentration $n = 1.3 \times 10^{19} \text{ cm}^{-3}$. We have added carrier concentration to the revised manuscript on **line 28, page 3**: "*From $m_{(e,h),j}^* = \hbar^2 k_j \frac{\partial k_j}{\partial E_{(C,V)}}$, we get the anisotropic reduced effective mass, where: $m_x^* = 0.12m_0$, $m_y^* = 0.32m_0$, $m_z^* = 0.10m_0$, m_0 is the electron mass*" and on **line 7, page 4**: "*We fitted the nano-FTIR spectra using a dipole model incorporating the effective dielectric constant $\langle \epsilon \rangle = \sqrt{\langle \epsilon_{plane} \rangle \epsilon_z}$ at a pump-probe delay $\tau = 200 \text{ fs}$ with a carrier concentration corresponding to $1.3 \times 10^{19} \text{ cm}^{-3}$ ".*

Comment 3-3: The nearly constant plasmon wavelength with varying BP thickness shown in Fig. 3 is interesting but also raises some question. I understand the argument about the finite penetration depth causing BP thicker than 100 nm to have the same transient plasmon wavelength. However, one can study BP films thinner than 100 nm, which should have homogeneous doping and will have more pronounced thickness dependence at e.g. 20 nm and 60 nm. Is the reason for lack of thinner BP samples in the current manuscript related to the surface degradation?

Response 3-3: The authors thank the referee for this question. We fabricated new samples, conducted relevant experiments, and added the content to Fig. S9.

In Fig. S9a-d, we selected four different BP slabs with thicknesses from 21 – 164 nm to explore the thickness-related polariton fringes. In Fig. S9e-h, we found that the

photo-induced plasmon fringes are only apparent in the BP slabs thicker than 102 nm, and the fringe contrast increases significantly with the increase in BP's thickness.

As the thickness of BP decreases, the fringe spacing in Fig. S9g (102 nm thick BP) reduces by about 13% compared with Fig. S9h (equivalent to 120 nm thick BP), which shows the BP's hyperbolic transient plasmon fringe spacing almost linearly decreases with the reduction in BP thickness, similar to the case of hBN phonon polaritons (*Science* 2014, 343, 1125). The cause of fringe vanishing in thin BP flakes may be twofold. One is that the oxidization rate of thin BPs becomes high (*Nat. Mater.* 2015, 14, 826–832), leading to an increasing plasmon propagation loss due to impurities scattering (Fig. S11; *Sci. Adv.* 2022, 8, eabn0627). The thinner the flake is, the stronger the oxidization effects. On the other hand, thin BP flakes exhibit a relatively weaker plasmonic response, leading to a decrease in the near-field amplitude signal of plasmon. Similar fringe contrast deduction with flake thickness also occurs in typical hyperbolic materials such as hBN and MoO₃, where the fringe contrast of polaritons decreases rapidly with decreasing thickness (*Adv. Mater.* 2018, 30, 1705318; *Adv. Mater.* 2019, 31, 1806603). Here, we cite the graph in *Adv. Mater.* 2018, 30, 1705318 to show similar results.

The thin BP samples degrade more rapidly in the atmospheric environment and do not last a long measurement time essential for dynamic near-field measurements. Therefore, we prefer thick samples exceeding 200 nm for studying the non-equilibrium plasmons in BP.

Fig. S9. Plasmons of different thin BP slabs. a) – d) Topography of BPs with a thickness of 21 – 164 nm. Scale bar: 1 μm . e) – h) Near-field amplitude images of BP with a thickness of 21 – 164 nm. The frequency region of the probe is $\nu/c = 850 - 1200 \text{ cm}^{-1}$.

[REDACTED]

Fig. R1. s-SNOM images of the α -MoO₃ flakes with different thicknesses. It is taken from *Adv. Mater.* 2018, 30, 1705318.

Comment 3-4: at page 7, second to last paragraph, the authors wrote "The low initial carrier density is attribute to the absence of polaritons fringes at $\tau=2$ ps." I think the authors meant the other way around, i.e. absence of fringe is attributed to low carrier density.

Response 3-4: We thank the reviewer for this suggestion and have corrected the corresponding errors in the text. We have replaced the "The low initial carrier density is attribute to the absence of polaritons fringes at $\tau=2$ ps" with "*At $\tau = 2$ ps, the absence of fringe is attributed to low initial carrier density*" on **line 29, page 7** in the revised manuscript.

REVIEWERS' COMMENTS

Reviewer #1 (Remarks to the Author):

I thank the authors for addressing my comments. I recommend this work for publication.

Reviewer #2 (Remarks to the Author):

This manuscript explores the active topological plasmonic dispersion in black phosphorus induced by ultrafast laser pulses. The study brings to light the distinctive hyperbolic polaritonic properties of black phosphorus, showcasing characteristics like remarkable long-range propagation, efficient edge excitations, and the coexistence of polaritons with varying relaxation times. The authors have carefully addressed the raised concerns, enhancing both the clarity of their explanations and the quality of the figures. With these improvements, I recommend its publication in Nature Communications.

Reviewer #3 (Remarks to the Author):

I have read the responses and the revised manuscript. The authors have addressed all my questions and comments. I only have a few additional suggestions before the publication of the manuscript in Nature Communications.

1. The revised Fig. 1h looks clear and compelling. To show the double-ring feature more quantitatively, the authors should consider plotting the radial line profiles of the pristine and non-equilibrium data in the same region in the Supplementary Information.
2. The relatively weak modulation of the plasmon wavelength with respect to the thickness is still surprising to me. The fluence dependence of the plasmon wavelength shown in Fig. S6 is important and the authors should consider comparing the extracted wavelength to the simulations in Fig. S6d.
3. The authors should provide some simulation to rule out the possible influence of the pump laser (1.55 um wavelength) on the observed fringes (around 1 um wavelength). For example, what is the dielectric constant of the top P_xO_y layer and the corresponding pump laser wavelength in the structure shown in Fig. S4?

Response Letter to Reviewers

Reviewer #1:

General comment: I thank the authors for addressing my comments. I recommend this work for publication.

General response: The authors greatly appreciate the referee's recognition of this work.

Reviewer #2:

General comment: This manuscript explores the active topological plasmonic dispersion in black phosphorus induced by ultrafast laser pulses. The study brings to light the distinctive hyperbolic polaritonic properties of black phosphorus, showcasing characteristics like remarkable long-range propagation, efficient edge excitations, and the coexistence of polaritons with varying relaxation times. The authors have carefully addressed the raised concerns, enhancing both the clarity of their explanations and the quality of the figures. With these improvements, I recommend its publication in Nature Communications.

General response: The authors are very grateful to the reviewer for the high evaluation of our work.

Reviewer #3:

General comment: I have read the responses and the revised manuscript. The authors have addressed all my questions and comments. I only have a few additional suggestions before the publication of the manuscript in Nature Communications.

General response: The authors thank the referee for valuable suggestions to improve this work. We have made revisions according to the referee's suggestions.

Comment 3-1: The revised Fig. 1h looks clear and compelling. To show the double-ring feature more quantitatively, the authors should consider plotting the radial line profiles of the pristine and non-equilibrium data in the same region in the

Supplementary Information.

Response 3-1: The authors thank the reviewer for this comment.

We have updated Supplementary Figure 8 in Supplementary Information. Supplementary Figure 8d shows the radial line profiles of Supplementary Figure 8a-b along the dashed arrow in Supplementary Figure 8b, which clearly exhibits the polariton fringe changes with pump fluence. The vertical dashed lines in Supplementary Figure 8d indicate the position of the edge of the gold disk. We have added the sentence: "Supplementary Figure 8d shows the radial line profiles of Supplementary Figure 8a-b, along the same direction of the dashed arrow in Supplementary Figure 8b. The profile line extracted at 0.5 mJ/cm² pumping power exhibits the polariton fringe while vanishing at 0 mJ/cm² pumping condition. The vertical black dashed line in Supplementary Figure 8d marked the position of the edge of the gold disk" in Supplementary Information, page 7, line 173.

Updated Supplementary Figure 8

Original Supplementary Figure 8

Comment 3-2: The relatively weak modulation of the plasmon wavelength with respect to the thickness is still surprising to me. The fluence dependence of the plasmon wavelength shown in Supplementary Figure 6 is important and the authors should consider comparing the extracted wavelength to the simulations in Supplementary Figure 6d.

Response 3-2: The authors thank the reviewer for this comment.

We have extracted polariton wavevectors in Supplementary Figure 6c and marked them as the cyan stars in Supplementary Figure 6d to show the center of polariton frequency. The relevant sentence: "The polariton wave vectors in Supplementary Figure 6c have been marked as cyan stars to show the center of polariton frequency." has been added in Supplementary Information, page 6, line 150.

Updated Supplementary Figure 6

Original Supplementary Figure 6

Comment 3-3: The authors should provide some simulation to rule out the possible influence of the pump laser (1.55 μm wavelength) on the observed fringes (around 1 μm wavelength). For example, what is the dielectric constant of the top P_xO_y layer and the corresponding pump laser wavelength in the structure shown in Supplementary Figure 4?

Response 3-3: The authors thank the reviewer for this comment.

We have simulated the propagation mode of $\text{Re}(E_z)$ at the pump light wavelength (1560 nm) in Supplementary Figure 4f. Using the approximated refractive index of BP (set as $\epsilon_{x,y,z} = 10.3+0.1i, 8.6+0.17i, 10.8+2.67i$) and Si ($\epsilon_{\text{Si}} = 12$) at pump light wavelength (*J. Phys. C* 1984, 17, 1839; *2D Mater.* 2022, 9, 015020), the electromagnetic field could not exhibit as a waveguide mode in BP and instead diffuses into Si quickly. In this work, a Ge filter (long pass) in the detection light path blocks the pump light in front of the detector in measurement to avoid the possible influence of the pump light. We have added the description: “Supplementary Figure 4f shows the simulated propagation mode of $\text{Re}(E_z)$ based on pump light wavelength (1560 nm). Using the approximated refractive index of BP (set as $\epsilon_{x,y,z} = 10.3+0.1i, 8.6+0.17i, 10.8+2.67i$) and Si ($\epsilon_{\text{Si}} = 12$) at pump light wavelength, the electromagnetic field could not exhibit as a waveguide mode in BP and instead diffuses into Si quickly. In this work, a Ge filter (long pass) in the detection light path blocks the pump light in front of the detector in measurement to avoid the possible influence of the pump light.” into Supplementary Information, page 4, line 106.

The dielectric constant of the top P_xO_y layer is set as 27 (*2D Mater.* 2022, 9, 015007), and the corresponding band gap is 7.2 eV, which is far larger than the pump energy $h\nu = 0.8$ eV (*Angew. Chem. Int. Ed.* 2016, 55, 1 – 6). We have added these parameters to the "Thickness-dependent polariton wavelength" part in Supplementary Information. We have added the description: “, the dielectric constant of P_xO_y and intrinsic BP is $\epsilon_{\text{P}_x\text{O}_y} = 27, \epsilon_{\infty,x} = 18, \epsilon_{\infty,y} = 14$ and $\epsilon_{\infty,z} = 9.75$, respectively. Since the band gap of the P_xO_y layer is 7.2 eV, which is far larger than pump energy $h\nu = 0.8$ eV, we do not need to consider the impact of the P_xO_y layer on the pump light. ” into

Updated Supplementary Figure 4

Original Supplementary Figure 4